# Identification of potent inhibitors of HDAC2 from herbal products for the treatment of colon cancer: Molecular docking, molecular dynamics simulation, MM/GBSA calculations, DFT studies, and pharmacokinetic analysis

**Madan Khanal**[1], **Arjun Acharya**[1], **Rajesh Maharjan**[1], **Kalpana Gyawali**[1], **Rameshwar Adhikari**[2,3], **Deependra Das Mulmi**[4], **Tika Ram Lamichhane**[1]*, **Hari Prasad Lamichhane**[1]

1 Central Department of Physics, Tribhuvan University, Kathmandu, Nepal, 2 Central Department of Chemistry, Tribhuvan University, Kathmandu, Nepal, 3 Research Center for Applied Science and Technology (RECAST), Tribhuvan University, Kathmandu, Nepal, 4 Nanomaterials Research Laboratory, Nepal Academy of Science and Technology (NAST), Lalitpur, Nepal

* tika.lamichhane@cdp.tu.edu.np

## Abstract

The histone deacetylase 2 (HDAC2), an enzyme involved in gene regulation, is a potent drug target for the treatment of colon cancer. Phytocompounds having anticancer properties show the ability to interact with HDAC2 enzyme. Among the compounds, docking scores of caffeic acid (CA) and *p*-coumaric acid (pCA) with HDAC2 showed good binding efficacy of -5.46 kcal/mol and -5.16 kcal/mol, respectively, with small inhibition constants. The higher binding efficacy of CA compared to pCA can be credited to the presence of an extra oxygen atom in the CA molecule, which forms an additional hydrogen bond with Tyr297. The HDAC2 in complex with these molecules was found to be stable by analyzing RMSD, RMSF, $R_g$, and SASA values obtained through MD simulations. Furthermore, CA and pCA exhibited low MM/GBSA free energies of -16.32 ± 2.62 kcal/mol and -17.01 ± 2.87 kcal/mol, respectively. The HOMO and LUMO energy gaps, dipole moments, global reactivity descriptor values, and MEP surfaces showed the reactivity of the molecules. The favourable physicochemical and pharmacokinetic properties, along with absence of toxicity of the molecules determined using ADMET analysis, suggested both the acids to be regarded as effective drugs in the treatment of colon cancer.

## Introduction

Colon cancer is one of the major causes of cancer death, resulting from the uncontrolled growth of cells in the colon or large intestine [1]. The cancerous growth can be controlled by inhibiting proper proteins [2]. Histone deacetylase 2 (HDAC2), among class I histone deacetylases (HDACs), is such an enzyme present in the histone proteins leading to the uncontrolled

**Data Availability Statement:** All relevant data are within the paper and its Supporting information files.

**Funding:** The author(s) received no specific funding for this work.

**Competing interests:** The authors declare no conflict of interest.

growth of colon cells [3]. The active site in HDAC2 consists of amino acid residues Gly154, Phe155, His183, Phe210, and Leu276 which contribute to the formation of lipophilic tube with the secondary structure of the protein as a loop region, while Tyr29, Met35, Phe114, and Leu144 play a role in forming the foot pocket with the secondary structure as a loop except for Phe114, which is an alpha-helix, nearby to the lipophilic tube [4]. The HDAC2 enzyme facilitates cell growth, infiltration, and spreading, while deactivating tumor suppressor genes via the deacetylation mechanism [5]. Inhibiting the function of HDACs presents an encouraging approach to decelerate the proliferation of cancerous cells. Certain HDAC inhibitors like suberanilohydroxamic acid (SAHA) and trichostatin A (TSA) have displayed encouraging outcomes to control cancer, though their success rate remains limited [2, 6]. Despite the global efforts to lower cancer rates, it has become the leading cause of death in recent years, and its incidence continues to rise [7]. The effective tackling of the disease to achieve significant progress in cancer treatment remains a challenge. This is mainly attributed to the considerable toxicity of existing drugs and the frequent development of resistance by tumor cells [8]. Consequently, there is a critical need to explore new, safe, and effective plant-based anticancer drugs.

Phenolic acids are commonly found in plant-derived foods and play crucial role in biological activities [9]. Caffeic acid (CA) (Fig 1a) and *p*-coumaric acid (pCA) (Fig 1b) are common phenolic compounds found in fruits and vegetables and show anticancer properties [10–13]. Both compounds have gained considerable interest due to their ring systems containing the hydroxyl and carboxyl groups [9, 14]. The number of hydroxyl (OH) substituents on the ring of phenolic acids have a significant impact on the observed antiproliferation and cytotoxicity against the cancer cell lines [15]. The study in HDAC inhibitory activity of cinnamic acid derivatives, including caffeic acid and *p*-coumaric acid, in colon cancer cells shows the deceleration of cancer cell growth [16, 17]. Additionally, the HDAC inhibitory activity is highlighted using phenolic-rich extracts derived from the rhizome of *Hydnophytum formicarum* Jack [18]. The anticancer efficacy of caffeic and coumaric acids is enhanced by diminishing the proliferation, adhesion, and migration of human lung (A549) and colon (HT29-D4) cancer cell lines [19]. These acids have also demonstrated inhibitory effects on telomerase reverse transcriptase (hTERT) with low expression of telomerase in the normal cells but high expression in tumor cells [20].

The computational methods, including DFT, molecular docking, MD simulations, and MM/GBSA, demonstrated that vanillic acid and piceatannol have strong binding affinities and stability with the human epidermal growth factor receptor 2 (HER2) target, highlighting their potential as effective anticancer properties [21]. The docking results indicate that the phenolic

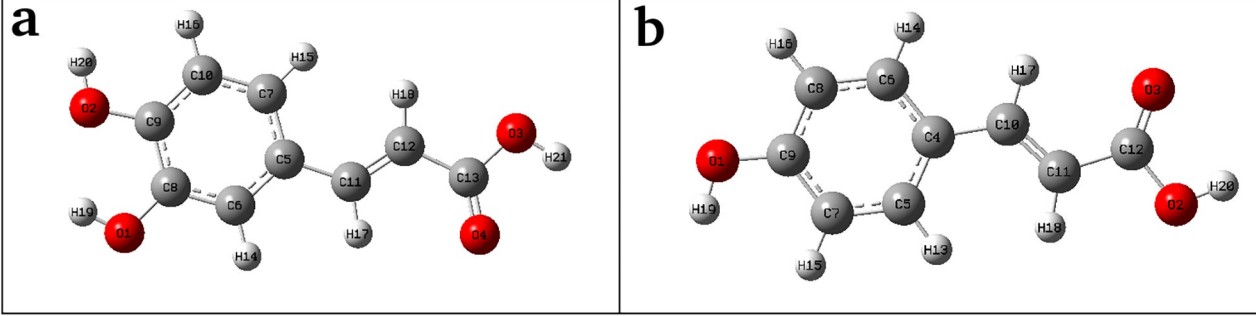

**Fig 1. Optimized structures using the DFT method at B3LYP/6–311++G(d,p) level of calculation.** (a) caffeic acid and (b) *p*-coumaric acid molecules.

acids present in *Moringa oleifera* leaves exhibit favorable docking energy values against the Bcl-2-associated X (BAX) protein, which promotes apoptosis and consequently reduces cancer cell viability [22]. The molecular docking and MD simulations investigates the binding modes, and ADME properties also gives drug-like nature of polyphenolic compounds from genus *Scrophularia* with aldose reductase (ALR2), identifying acacetin as a stable and effective ligand that exhibits anticancer activity [23].

Molecular docking and molecular dynamics (MD) simulations play a vital role in the drug discovery process. These approaches provide valuable insights into the interaction of repurposed compounds with specified targets, elucidating their potential for additional therapeutic applications [24]. Information regarding the structural, electronic, and spectroscopic characteristics of compounds can be acquired using the density functional theory (DFT) approach [25–28]. The consideration of ADME parameters, alongside the assessment of toxicity of potential drug candidates, plays a crucial role in the drug development process [29–31].

This research aims to identify promising phytocompounds with anticancer properties, followed by exploring the inhibitory effects of the selected compounds CA and pCA against enzyme HDAC2 using molecular docking, MD simulations, and post-MD free energy calculations. Additionally, it seeks to investigate the quantum chemical properties of the molecules through DFT analysis and evaluate their ADMET profiles for potential colon cancer treatment applications.

## Materials and methods

### Molecular docking

The crystallographic structure of HDAC2 complexed with N-(4-aminobiphenyl-3-yl)benzamide at resolution 2.05 Å (PDB ID: 3MAX) was downloaded from the online server RCSB Protein Data Bank [32]. The structure was refined using the online server SWISS-MODEL [3]. During refinement, the template was found using HHblits with a global model quality estimate (GMQE) score of 0.99, derived from X-ray diffraction at a resolution of 1.66 Å, representing the monomeric form of the protein structure. In RCSB PDB, residues were numbered from Ala12 to Leu378, while in SWISS-MODEL, after refinement, residues were numbered from Ala1 to Leu367. Despite this difference in numbering, the total number of residues in the protein remained 367, and the sequence was consistent without any missing residues even after refinement. The structure was further refined using the AutoDockTools [33] by removing water molecules, adding polar hydrogen atoms, and assigning Kollman charges to all protein atoms.

Seventeen bioactive molecules having anticancer properties were chosen from the literature (S1 Table) [18, 34–40] and three dimensional chemical structures were obtained from PubChem [41]. The binding energies of phytochemicals in the active sites of receptor enzyme HDAC2 were virtually screened using PyRx [42]. Two compounds showing the least binding energies from above screening were docked with HDAC2 using AutoGrid4.2 and Auto-Dock4.2 of AutoDockTools within the grid points 60 × 60 × 60 along x, y, and z directions, respectively, at grid center x = 67.987, y = 31.384, and z = 3.889 with grid spacing 0.375 Å. After molecular docking, the most stable receptor-ligand complexes were chosen among the 100 poses obtained by analyzing binding affinity, number of hydrogen bonds, conformations in the cluster, inhibition constant, and dipole moment. The native co-crystallized ligand of HDAC2 was re-docked into its binding site and RMSD was calculated using the DockRMSD program [43] for the validation of docking procedures. The complexes were visually inspected in PyMOL 2.5.2 [44] and BIOVIA Discovery Studio Visualizer V21.1.0.20298 [45]. The

interactions between the protein and the compounds were also analyzed using the LigPlot + V.2.2 program [46].

## Molecular dynamics simulations

MD simulations were performed for the apo form of HDAC2 and the best-docked protein-ligand complexes obtained from molecular docking using GROMACS 2019.6 software package [47]. The chemistry at Harvard macromolecular mechanics (CHARMM) force field with CHARMM27 parameter set [48] was employed solvating the complexes using the transferable intermolecular potential with a 3 points (TIP3P) water model throughout the simulation [49]. The energy minimization process was performed using the steepest descent method and $Na^+$ and $Cl^-$ ions at the concentration of 0.1 M were added for neutralization. All essential topology files, including CHARMM parameter files, were generated using the SwissParam server [50]. MD simulations of the complex were conducted for total simulation time of 200 nanoseconds (ns), collecting snapshots at intervals of 100 picoseconds (ps) from 0 ns to 200 ns. The root-mean-square deviation (RMSD), root-mean-square fluctuation (RMSF), radius of gyration ($R_g$), solvent accessible surface area (SASA), and hydrogen bond interactions (H-bond) were analyzed for the apo (the protein without a bound ligand) and complex forms to test the conformational stability [51]. The interactions in the complexes were visually inspected in PyMOL 2.5.2, BIOVIA Discovery Studio Visualizer V21.1.0.20298, and LigPlot+ V.2.2 program.

## MM/GBSA approach

The average binding free energies were calculated using MM/GBSA approach after MD simulations to access the interaction of the complexes, along with its standard deviation. The gmx_MMPBSA and gmx_MMPBSA_ana tools were used to calculate the binding free energy with statistical precision of 1 kJ/mol in the results and generate graphical visualizations of the complexes obtained from MD simulations [52, 53]. Molecular mechanics generalized Born surface area (MM/GBSA) approach was applied to compute the binding free energies ($\Delta G_{binding}$) of the complexes [54]. The binding free energy calculations can be obtained by using the Eq (1) [55],

$$\Delta G_{binding} = G_{complex} - (G_{receptor} + G_{ligand}) \tag{1}$$

where, $\Delta G_{binding}$ is the binding free energy of ligand-protein complex, $G_{complex}$ is free energy of the complex, $G_{receptor}$ and $G_{ligand}$ are the free energies of the unbound receptor and ligand, respectively.

The contributions of different interactions are expressed as [56],

$$\Delta G_{binding} = \Delta E_{MM} + \Delta G_{solv} - T\Delta S \tag{2}$$

in which

$$\Delta E_{MM} = \Delta E_{int} + \Delta E_{ele} + \Delta E_{vdW} \tag{3}$$

$$\Delta G_{solv} = \Delta G_{PB} + \Delta G_{SA} \tag{4}$$

where, gas-phase interaction energy between ligand and protein ($\Delta E_{MM}$) is the sum of internal energy ($\Delta E_{int}$), electrostatic interaction ($\Delta E_{ele}$), and van der Waals interaction energies ($\Delta E_{vdW}$). The solvation free energy ($\Delta G_{solv}$) is estimated as a sum of the polar contribution to the solvation free energy ($\Delta G_{PB}$), calculated using the Poisson-Boltzmann implicit solvent

model, and the non-polar contribution is calculated based on the solvent accessible surface area ($\Delta G_{SA}$). The Poisson-Boltzmann implicit solvent model estimates the polar solvation free energy by treating the solvent as a continuous, uniform medium without explicit solvent molecules, calculating the interactions between solute atoms and the implicit solvent. The changes in conformational entropy upon ligand binding (T$\Delta$S) were neglected due to the heavy computational cost.

### DFT calculations

The Gaussian input files of the molecules were generated with the GaussView 6 program [57] and quantum chemical calculations were performed using Gaussian 16W package [58] at the DFT/B3LYP/6–311++G(d,p) level of calculations [59, 60]. The frontier molecular orbitals (FMOs), energy gaps, and reactivity descriptors were calculated using the time-dependent density functional theory (TD-DFT) method [61]. The density of states (DOS) plots were also generated using GaussSum software [62]. The NBO version 3.1 calculations for natural bonding orbital (NBO) analysis were conducted using the DFT method at the same level of calculations [63, 64].

### ADMET calculations

The absorption, distribution, metabolism, excretion and toxicity (ADMET) of the molecules were calculated using admetSAR 2.0 web server [65], SwissADME website [66] and ProTox-II [67]. These properties include topological polar surface area (TPSA), water solubility and Lipinski's rule of 5 [30].

## Results and discussion

### Molecular docking

Among the seventeen phytochemicals having anticancer properties, the caffeic acid (Compound ID 689043) and *p*-coumaric acid (Compound ID 637542) molecules show high binding efficacy with HDAC2 (S1 Table). The active amino acids within the binding pocket of the protein HDAC2 are Tyr29, Met35, Phe114, Leu144, His145, His146, Gly154, Cys156 and Tyr308 [3, 4] and after refinement of the protein, the numbers of the above active residues changes to Tyr18, Met24, Phe103, Leu133, His134, His135, Gly143, Cys145 and Tyr297, respectively. The low RMSD value (0.448 Å) (S1 Fig) of initial co-ordinates and the generated co-ordinates of native ligand indicates the reliability of the docking method [68].

It is observed that HDAC2 enzyme and CA molecule form 5 H-bonds with a distance of 2.57 Å, 2.82 Å, 2.85 Å, 3.06 Å, and 3.12 Å. Amino acid residues Tyr18, Met24, Gly132, Leu133, Gly143, Phe144, Cys145, Gly294, and Gly295 are involved in the hydrophobic interactions with CA and Cys145 is involved in $\pi$-alkyl interaction (Table 2 and Fig 2c). The total binding energy of CA molecule with HDAC2 enzyme is -5.46 kcal/mol with an inhibition constant 99.10 $\mu$M (Table 1). It is observed that HDAC2 enzyme and pCA molecule form 4 H-bonds with a distance of 2.59 Å, 2.96 Å, 2.99 Å, and 3.02 Å. Amino acid residues Tyr18, Met24, Gly132, Leu133, Gly143, Phe144, Cys145, Gly294, Gly295, and Tyr297 are involved in the hydrophobic interactions and Cys145 is involved in $\pi$-alkyl interaction (Table 2 and Fig 2d). The total binding energy of pCA molecule with HDAC2 enzyme is -5.16 kcal/mol with an inhibition constant 166.20 $\mu$M (Table 1). The interactions between the various residues of protein HDAC2 and the compounds CA and pCA are shown in S2 Fig. The binding energy, interacting amino acids, and hydrogen bond formation collectively indicate that CA has higher binding efficacy with HDAC2 compared to pCA.

**Table 1. Molecular docking results for CA-HDAC2 and pCA-HDAC2 complexes.**

| Parameters | CA-HDAC2 complex | pCA-HDAC2 complex |
|---|---|---|
| Final Intermolecular Energy (vdW + H-bond + Desolvation Energy + Electrostatic Energy)(kcal/mol) | -6.95 | -6.35 |
| Final Total Internal Energy (kcal/mol) | -0.24 | -0.07 |
| Torsional Free Energy (kcal/mol) | 1.49 | 1.19 |
| Unbound System's Energy (kcal/mol) | -0.24 | -0.07 |
| Estimated Free Energy of Binding (kcal/mol) | -5.46 | -5.16 |
| Inhibition Constant, $K_i$ ($\mu M$) | 99.10 | 166.20 |

**Table 2. Interactions of HDAC2 protein residues with caffeic acid and *p*-coumaric acid molecules.**

| Ligand | Target protein | Binding residues | Atoms | Bond length (Å) | Interactions |
|---|---|---|---|---|---|
| CA | HDAC2 | His134 | NE2-O4 | 3.12 | H-bonds |
| | | His135 | NE1-O4 | 3.06 | H-bonds |
| | | Asp170 | OD1-O4 | 2.57 | H-bonds |
| | | His172 | ND1-O4 | 2.85 | H-bonds |
| | | Tyr297 | OH-O1 | 2.82 | H-bonds |
| | | Tyr18, Met24, Leu133, | | | |
| | | Phe144, Gly143, | | | |
| | | Gly294, Gly295, | | | Non-bonded |
| | | Cys145 | | | |
| pCA | HDAC2 | His134 | NE2-O1 | 3.02 | H-bonds |
| | | His135 | NE2-O1 | 2.99 | H-bonds |
| | | Asp170 | OD1-O1 | 2.59 | H-bonds |
| | | His172 | ND1-O1 | 2.96 | H-bonds |
| | | Tyr18, Met24, Gly132, | | | |
| | | Leu133, Phy144, Gly295, | | | Non-bonded |
| | | Cys145 | | | |

## Molecular dynamics simulations

The RMSD analysis offers insights into the extent of structural deviation and conformational stability observed throughout the MD simulation period [29]. The time dependent RMSD and the RMSD relative frequency graphs are plotted to illustrate the fluctuations in the backbone structure of the protein in apo form and the protein complex involving the ligands CA and pCA with HDAC2 (Fig 3a and 3b). The average RMSD values for the backbone of apo form, CA-HDAC2, and pCA-HDAC2 complexes are observed to be 1.68 Å, 1.69 Å and 2.19 Å, respectively. The small RMSD values of the backbone atoms of protein and complexes indicate that HDAC2 is quite stable throughout the simulation. The average RMSD values of the heavy atoms of the ligand relative to the protein backbone in the CA-HDAC2 and pCA-HDAC2 complexes are 2.79 Å and 4.08 Å, respectively (S3 Fig). Within the time range of 60 to 100 ns, CA shows lower RMSD, during which Tyr18 contributes higher binding affinity. After 100 ns, pCA shows increased RMSD, during which Met24 and Cys145 demonstrate lower binding affinities (S4 Fig).

The packing and stability of the protein structure is analyzed by calculating the $R_g$ values with its relative frequency (Fig 3c and 3d). The presence of CA and pCA in the system with HDAC2 may continuously challenge the compactness of the HDAC2 stability. The average $R_g$

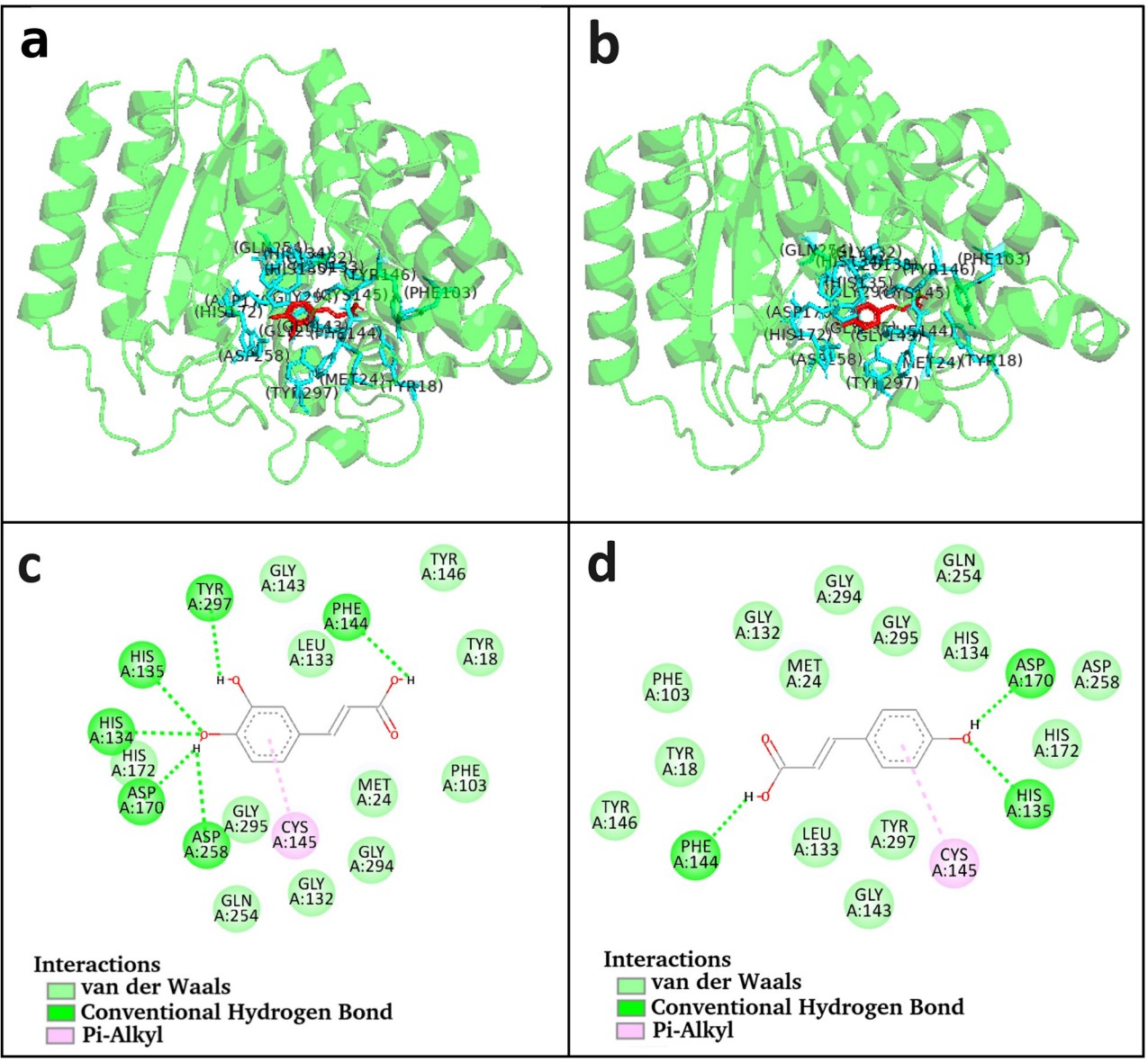

**Fig 2. Representation of molecular docking.** (a) CA-HDAC2 complex and (b) pCA-HDAC2 complex structures having ligand (red) and active site residues (cyan) visualized with PyMOL. (c) Interactions of CA and (d) pCA with active amino acid residues of HDAC2 using BIOVIA Discovery Studio software.

values of $C_\alpha$ atoms for the apo form (19.88 ± 0.06 Å), CA-HDAC2 complex (19.87 ± 0.07 Å), and the pCA-HDAC2 complex (19.88 ± 0.09 Å) are nearly identical. The HDAC2 protein is also significantly stable in the presence of CA and pCA molecules during the simulation period with low average values of $R_g$.

The SASA values of both molecules are computed to determine the accessibility of solvent molecules on the protein's surface and are then plotted in the Fig 3e and 3f. The average SASA values of the apo form (167.37 ± 2.81 nm$^2$), CA-HDAC2 complex (166.42 ± 2.86 nm$^2$) and pCA-HDAC2 complex (165.98 ± 2.54 nm$^2$) are stable during 200 ns MD simulations. The slight decrease in SASA values in the complexes implies a subtle tightening and enhanced stability of the protein-ligand systems [69].

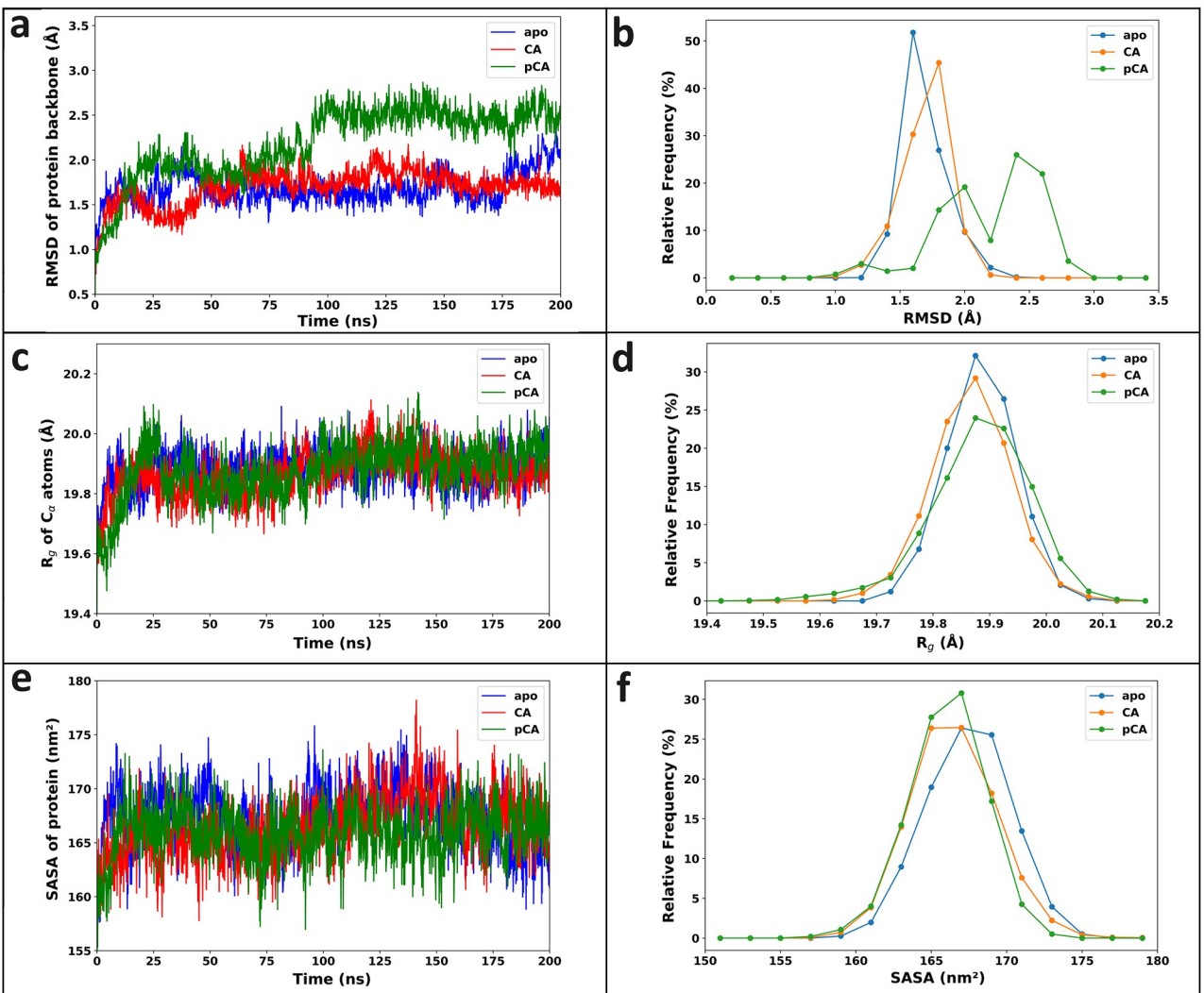

**Fig 3. Insights into molecular dynamics simulations.** (a) RMSD of the backbone atoms of HDAC2 and (b) its relative frequency, (c) $R_g$ plot and (d) its corresponding relative frequency, (e) average SASA of protein and (f) its relative frequency. The apo form of HDAC2 (blue), the CA-HDAC2 complex (red), and the pCA-HDAC2 complex (green) undergo variations during 200 ns MD simulations.

The calculation of RMSF provides details of the fluctuations in amino acid residues induced by the binding of molecule to protein [70]. The average value of RMSF for apo form is 0.75 ± 0.49 Å and for CA-HDAC2 and pCA-HDAC2 complexes are 0.83 ± 0.38 Å and 0.90 ± 0.56 Å, respectively. The average RMSF values are almost similar for CA-HDAC2 and pCA-HDAC2 complexes in comparison with apo form except few fluctuations. The maximum fluctuation in RMSF for both complexes and apo form is occurred at the residues 198, 199, and 200 (Fig 4). Again, among the active residues, Asp93 residue fluctuates with the maximum RMSF value for both the complexes as well as apo form (Table 3). The mentioned residues lie in loops and surface-exposed regions of the protein, which are generally more flexible [71].

In the CA-HDAC2 complex, the highest number of H-bonds formed during MD simulations is 4, with many conformations showing 2 H-bonds. In pCA-HDAC2 complex, the highest number of H-bonds formed is 3. Majority of conformations show 2 H-bonds during

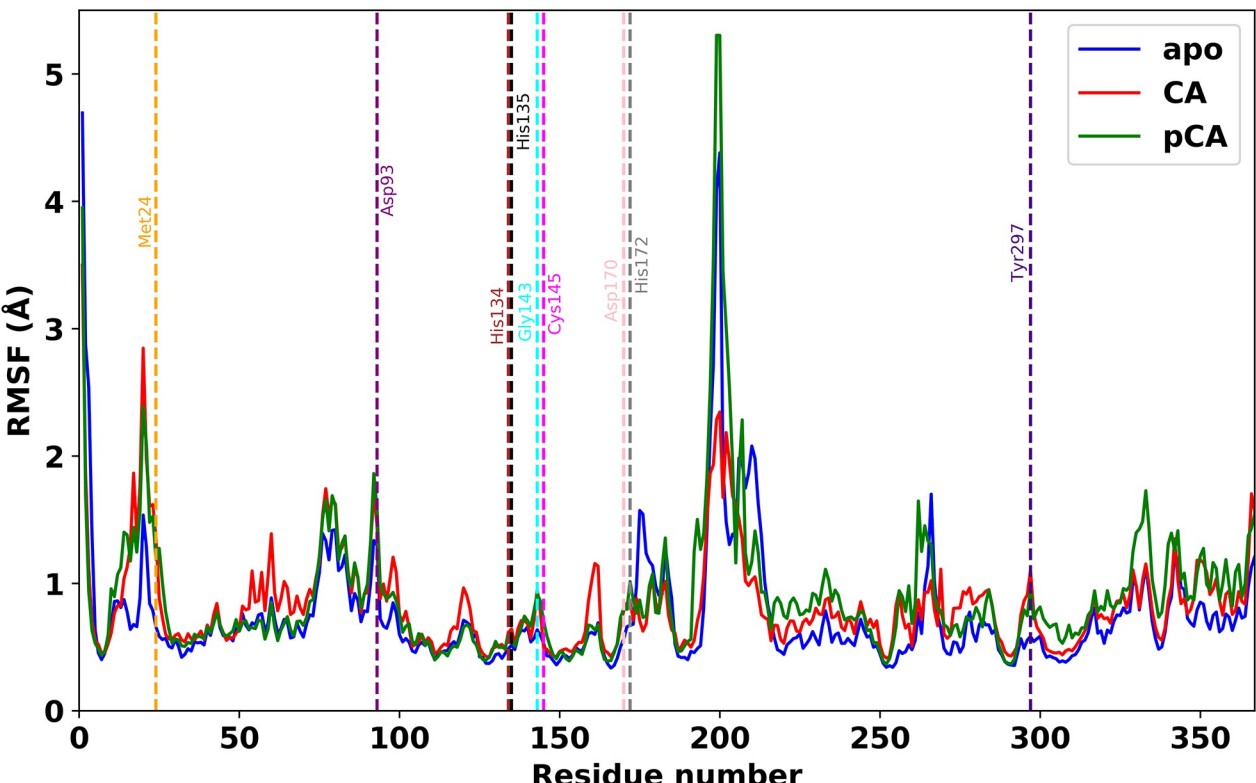

**Fig 4. RMSF of amino acid residues (C$_\alpha$ atoms) in apo form, CA-HDAC2 and pCA-HDAC2 complexes during 200 ns equilibration simulations.** The vertical dotted lines indicate the residues in the ligand binding pocket.

whole MD simulations (S5 Fig). These nonzero number of H-bonds (fluctuating around 2) between protein and ligand systems facilitates the formation of stable protein-ligand complex [72, 73].

The interacting amino acids with both compounds after molecular docking (Fig 2c and 2d) are almost the same during different frames and after the 2000$^{th}$ frame MD simulations (S6 Fig). This shows the correspondence between the docked compounds and the conformations obtained from MD simulations and the crystal structure, illustrating their consistency.

**Table 3. RMSF values of active amino acids in apo form, CA-HDAC2, and pCA-HDAC2 complexes during 200 ns equilibration run.**

| Active amino acids | RMSF (Å) | | |
|---|---|---|---|
| | apo form | CA-HDAC2 complex | pCA-HDAC2 complex |
| Met24 | 0.66 | 1.35 | 1.19 |
| Asp93 | 1.30 | 1.46 | 1.59 |
| His134 | 0.49 | 0.62 | 0.61 |
| His135 | 0.52 | 0.64 | 0.56 |
| Gly143 | 0.64 | 0.95 | 0.92 |
| Cys145 | 0.53 | 0.49 | 0.70 |
| Asp170 | 0.56 | 0.79 | 0.80 |
| His172 | 0.67 | 0.87 | 1.01 |
| Tyr297 | 0.58 | 1.10 | 0.96 |

## MM/GBSA calculations

The van der Waals energy (vdW), electrostatic energy (EEL), polar solvation energy (EGB), and non-polar solvation energy (ESURF) with total binding free energy ($\Delta G_{binding}$) for both complexes are presented in Fig 5a and 5b. The $\Delta G_{binding}$ values for CA-HDAC2 and pCA-HDAC2 complexes are -16.32 ± 2.62 kcal/mol and -17.01 ± 2.87 kcal/mol, respectively,indicating good binding scores (Table 4). The 162[nd] and 1439[th] frames of the MD simulations give the lowest total binding free energies for CA-HDAC2 complex (-24.54 kcal/mol) and pCA-HDAC2 complex (-24.29 kcal/mol), respectively (Fig 5c and 5d). Within the active residues of HDAC2, Met24 has the lowest binding energy of -2.05 ± 0.85 kcal/mol in CA-HDAC2 complex and Tyr18 has the lowest binding energy of -1.95 ± 0.66 kcal/mol in pCA-HDAC2 complex (Fig 5e and 5f). The binding free energies of the complexes reveal that both the selected bioactive molecules show good interaction with the HDAC2 enzyme.

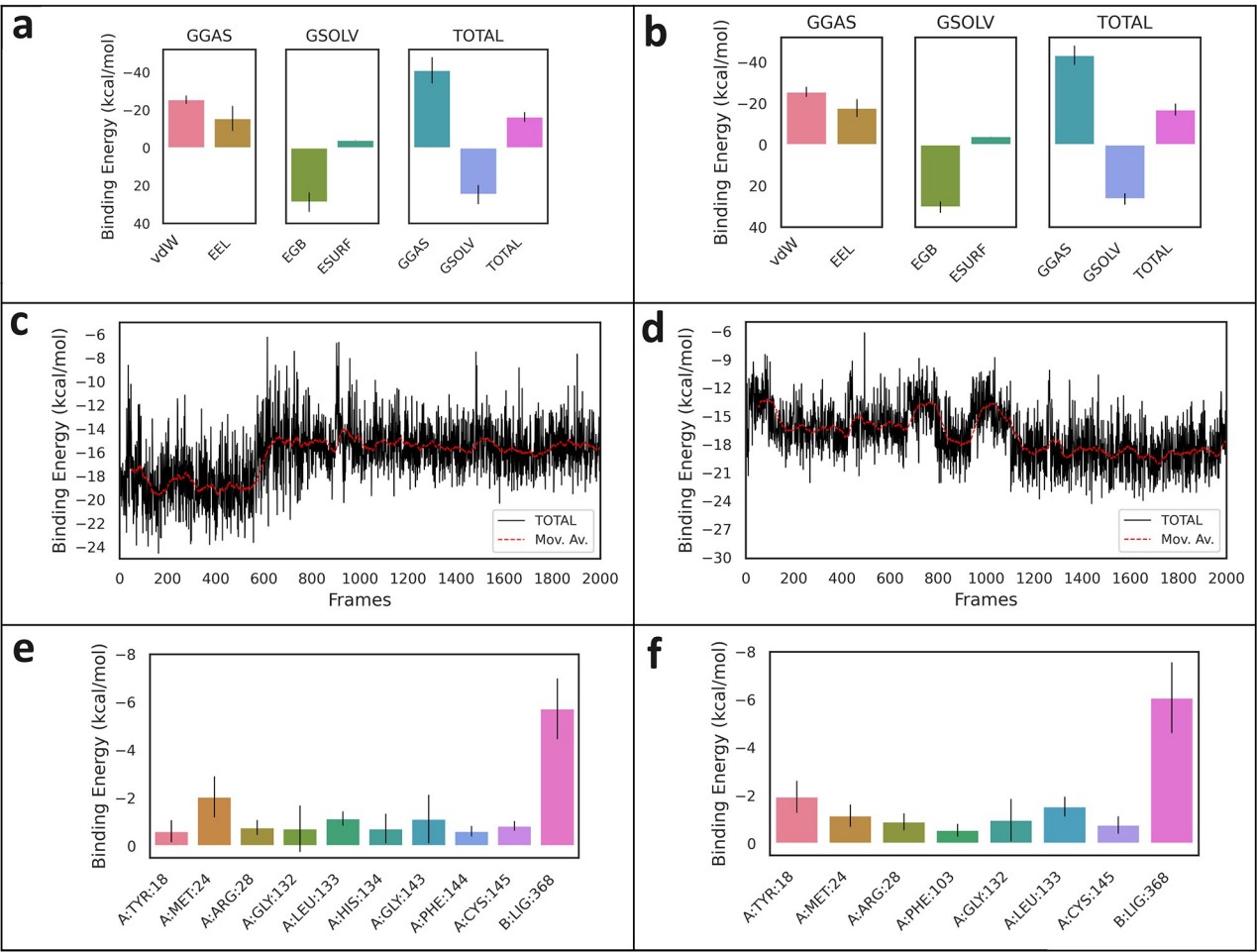

**Fig 5. Illustration of MM/GBSA analysis conducted after MD simulations.** (a) The influences of vdW forces, electrostatic interactions, and solvation energy contributions to MM/GBSA binding free energy for CA-HDAC2 and (b) for pCA-HDAC2 complexes; (c) time evolution changes of total binding free energy with moving average (Mov. Av.) for CA-HDAC2 and (d) for pCA-HDAC2 complexes; (e) partial contribution of active amino acids and ligands to the binding free energy of CA-HDAC2 and (f) pCA-HDAC2 complexes. Vertical lines in bars represent the standard deviations.

**Table 4. Calculated binding free energies by the MM/GBSA method for CA-HDAC2 and pCA-HDAC2 complexes.**

| Energy Components | CA-HDAC2 complex (kcal/mol) | pCA-HDAC2 complex (kcal/mol) |
|---|---|---|
| Van der Waals Energy | -25.50 ± 2.16 | -25.58 ± 2.44 |
| Electrostatic Energy | -15.56 ± 6.65 | -17.76 ± 4.43 |
| Polar Solvation Energy | 24.73 ± 5.07 | 26.32 ± 2.72 |
| Non-polar Solvation Energy | -4.01 ± 0.11 | -3.98 ± 0.10 |
| Estimated Binding Energy | -16.32 ± 2.62 | -17.01 ± 2.87 |

## DFT analysis

**Geometry optimization.**   The optimized molecular geometries of the CA and pCA molecules are shown in Fig 1a and 1b, respectively. The geometrical parameters such as calculated total energy (global minimum energy), dipole moment, the RMS Cartesian force and the maximum Cartesian force of the compounds are shown in Table 5. The global minimum energy for CA (-648.73 Hartrees) is less than that for pCA (-573.47 Hartrees) which indicates that CA is more stable than pCA. The small values of the root mean square (RMS) Cartesian force and maximum Cartesian force (Table 5) suggest that both molecules secure stable geometries [74]. The dipole moment indicates how the charges are distributed within a molecule which provides insight into the movement of charge across its structure [75]. The dipole moment for CA and pCA are 5.09547 Debye and 3.72153 Debye, respectively. This reflects stronger intermolecular interactions of CA molecule promoting the formation of more hydrogen bonds [76].

**Frontier molecular orbitals.**   The value of the energy difference between highest occupied molecular orbital (HOMO) and lowest unoccupied molecular orbital (LUMO) provides the reactivity of chemical compound [77]. The energies of six significant FMOs, specifically HOMO—2, HOMO—1, HOMO, LUMO, LUMO + 1, and LUMO + 2 are calculated using the TD-DFT method. The related plots show that the orbitals are localized mostly on the benzene ring (Fig 6a and 6b). The green colour represents the negative phase, the red colour represents the positive phase, and this aspect is clearly explained in the density of states (DOS) spectra of CA and pCA molecules as DOS spectrum characterizes the energy levels per unit energy (Fig 7a and 7b). The energy gaps between HOMO and LUMO orbitals are observed to be 4.16 eV and 4.27 eV for CA and pCA molecules, respectively, and the calculated values are consistent with those found through the DOS spectra. The small energy gap facilitates the flow of electrons, making the CA molecule is softer and more reactive than the pCA molecule.

**Global reactivity descriptors.**   The global chemical reactivity descriptors, electron affinity (A), ionization potential (I), chemical hardness ($\eta$), chemical softness ($\beta$), electronic chemical potential ($\mu$) and global electrophilicity index ($\omega$) are related by Eqs (5)–(8) as HOMO is directly related to ionization potential ($I = -E_{HOMO}$) and LUMO is related to electron affinity ($A = -E_{LUMO}$) [75, 78].

$$\eta = \frac{1}{2}(I - A) \tag{5}$$

**Table 5. Computed total energy, dipole moment, root mean square Cartesian force, and maximum Cartesian force of caffeic acid and p-coumaric acid.**

| Molecules | Total Energy (E) Hartrees | Dipole Moment ($\mu$) Debye | RMS Cartesian Force (Hartrees/Bohr) | Maximum Cartesian Force (Hartrees/Bohr) |
|---|---|---|---|---|
| CA | -648.73 | 5.09547 | $5.2435 \times 10^{-5}$ | $20.7555 \times 10^{-5}$ |
| pCA | -573.47 | 3.72153 | $5.2000 \times 10^{-8}$ | $12.5000 \times 10^{-8}$ |

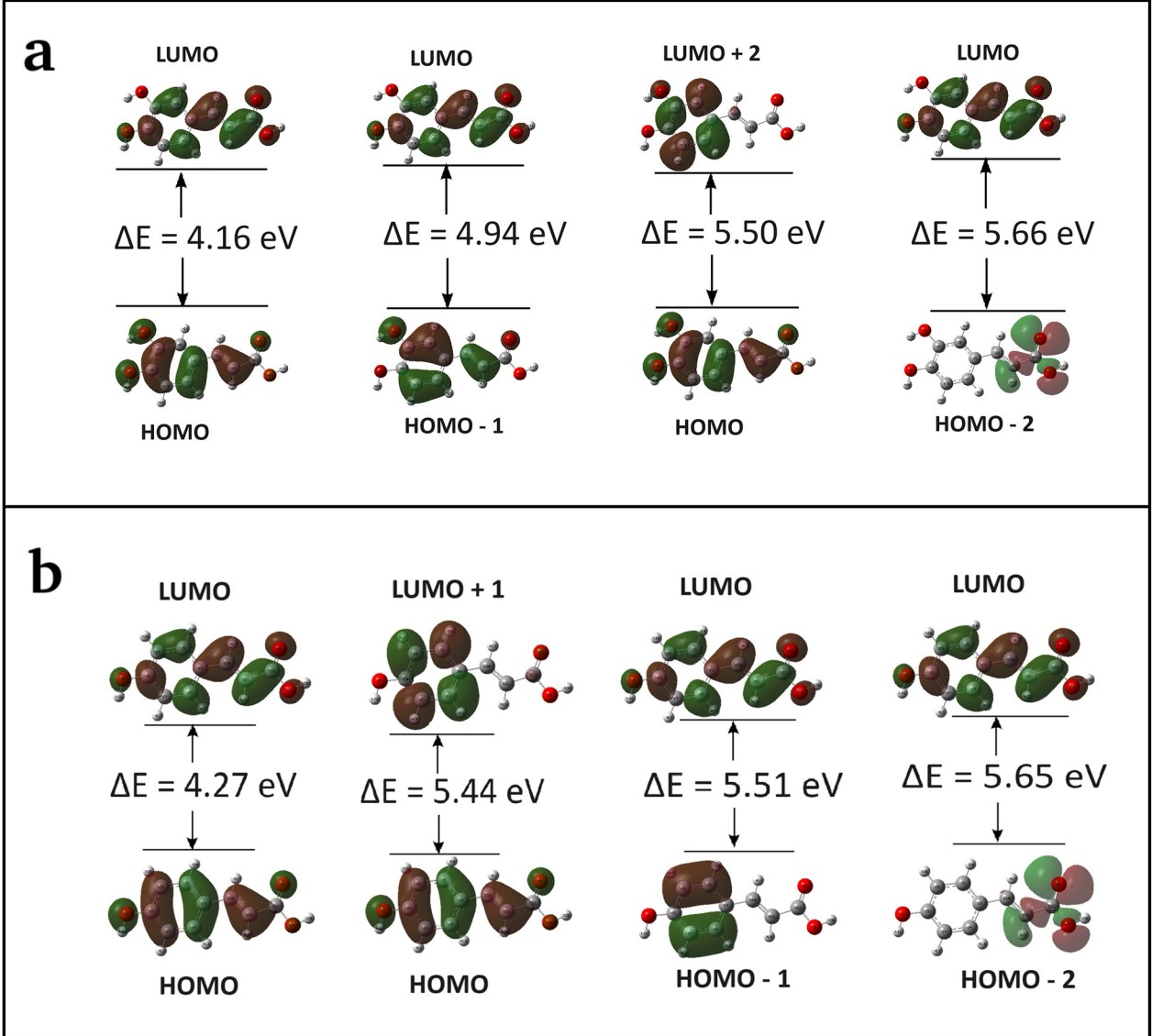

**Fig 6. Schematic diagrams illustrating energy gap of frontier molecular orbitals.** (a) caffeic acid and (b) *p*-coumaric acid molecules.

$$\beta = \frac{1}{\eta} \tag{6}$$

$$\mu = -\frac{1}{2}(I + A) \tag{7}$$

$$\omega = \frac{\mu^2}{2\eta} \tag{8}$$

Table 6 summarizes global reactivity descriptors of the molecules. The HOMO and LUMO energy gaps for the molecules CA and pCA are obtained as 4.16 eV and 4.27 eV, respectively. These small values of energy gap suggest the molecules are chemically reactive and involve in

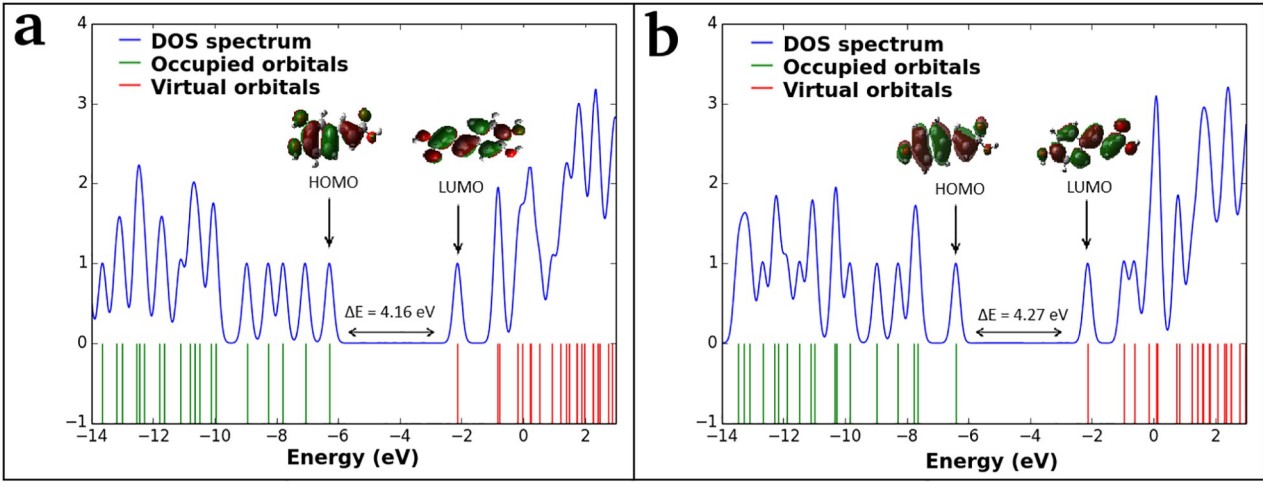

**Fig 7. Density of states spectra.** (a) caffeic acid and (b) *p*-coumaric acid molecules.

charge transfer interaction within the molecules [28]. The electrophilicity indices, 4.26 eV for CA and 4.28 eV for pCA along with electronic chemical potential, suggest the presence of significant electrophile in the molecules. The calculated electrophilicity index provides insight into the biological activity of CA and pCA molecules. The chemical hardness ($\eta$) reflects the ability of charge transfer inside the molecules. Low values of electronic chemical potential ($\mu$) (-4.21 eV for CA and -4.28 eV for pCA) indicate that both the molecules are capable to donate electrons.

**Molecular electrostatic potential.** The molecular reactive properties and intermolecular interactions can be illustrated using molecular electrostatic potential (MEP) diagram. The reactive electrophilic and nucleophilic sites aid in forming the hydrogen bonds [79]. The MEP diagram gives negative, neutral, and positive electrostatic potential regions by colour grading. The red colour represents the most electronegative electrostatic potential (strong attraction), the blue colour represents the most electropositive potential (strong repulsion), and green colour indicates the regions of zero potential. The maximum positive regions are located on the hydrogen H16, H19, H20, H21 atoms for the CA molecule and H15, H19, H20 atoms for the pCA molecule indicating possible sites for electrophilic attack. The oxygen atoms O4 in CA and O3 in pCA have maximum negative charges, indicating the nucleophilic sites (Fig 8a and 8b).

**Table 6. Global reactivity descriptors of caffeic acid and *p*-coumaric acid molecules.**

| Parameters | CA | pCA |
|---|---|---|
| $E_{HOMO}$ | -6.29 eV | -6.41 eV |
| $E_{LUMO}$ | -2.13 eV | -2.14 eV |
| $E_{LUMO}$—$E_{HOMO}$ | 4.16 eV | 4.27 eV |
| Electron affinity (A) | 2.13 eV | 2.14 eV |
| Ionization potential energy (I) | 6.29 eV | 6.41 eV |
| Chemical hardness ($\eta$) | 2.08 eV | 2.14 eV |
| Chemical softness ($\beta$) | 0.48 $(eV)^{-1}$ | 0.47 $(eV)^{-1}$ |
| Electronic chemical potential ($\mu$) | -4.21 eV | -4.28 eV |
| Global electrophilicity index ($\omega$) | 4.26 eV | 4.28 eV |

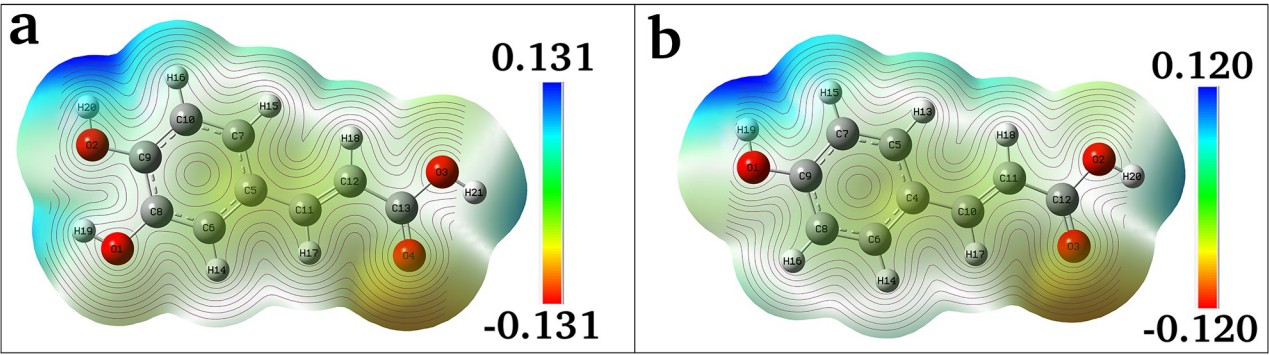

**Fig 8. Molecular electrostatic potential diagrams.** (a) caffeic acid and (b) *p*-coumaric acid molecules with contour lines indicating the possible sites for nucleophilic and electrophilic attacks.

**Absorption analysis.** The ultraviolet-visible (UV-Vis) spectra of the compounds are obtained to explain charge transfer, absorption properties, and excitation energies of the compounds. The wavelengths and oscillator strengths for the first excited state are higher than the other states for both molecules (Table 7). The calculated wavelengths of maximum absorption ($\lambda_{max}$) for both CA (322 nm, 284 nm, and 277 nm) and pCA (302 nm, 276 nm, and 273 nm) for dominant transitions are in good agreement with the experimental $\lambda_{max}$ for CA (327 nm) and pCA (288 nm), respectively. The wavelength 322 nm for CA and 302 nm for pCA are responsible for the majority of the formation of the absorption band (Fig 9a and 9b).

**Mulliken and natural charges.** Mulliken and natural charge calculations are commonly employed in quantum chemical calculations because they play a crucial role in determining the molecular polarizability, electronic structure, dipole moment and various other attributes of the molecule [76]. The atomic Mulliken charges and natural charges assigned to the atoms of both compounds reveal negative charges on oxygen atoms and positive charges on hydrogen atoms, while carbon atoms exhibit either negative or positive charges (Table 8). The maximum positive Mulliken charges are detected on C5 in the CA compound and C4 in the pCA compound, while the most negative charges were located on C7 for CA and C5 for pCA, respectively. The greatest positive natural charges are observed on C13 in the CA compound and C12 in the pCA compound, while the most negative natural charges are noticed on O2 for both CA and pCA (Fig 10a and 10b).

Regarding the hydrogen atoms, the majority of the Mulliken charge is concentrated on hydrogen atom H21 in CA and H20 in pCA. Furthermore, the charge on O4 surpasses that observed on O3, O2, and O1 atoms in CA, while the charge on O3 exceeds the values observed on O2 and O1 atoms in pCA. These findings indicate the existence of hydrogen bonds. Since

**Table 7. Calculated wavelengths of maximum absorption ($\lambda_{max}$) with major contributions of orbitals and oscillator strengths compared to experimental wavelengths of maximum absorption ($\lambda_{max}$) of caffeic acid and *p*-coumaric acid molecules.**

| Molecule | Calculated $\lambda_{max}$ (nm) | Oscillator Strength (f) | Orbital description for major contributions | Experimental $\lambda_{max}$ (nm) |
|---|---|---|---|---|
| CA | 322 | 0.28 | H→L (90%) | |
| | 284 | 0.24 | H-1→L (67%); H→L+2 (22%) | 327 [80] |
| | 277 | 0.00 | H-2→L (96%) | |
| pCA | 302 | 0.64 | H→L (97%) | |
| | 276 | 0.00 | H-2→L (96%) | 288 [81] |
| | 273 | 0.02 | H-1→L (48%); H→L+1 (50%) | |

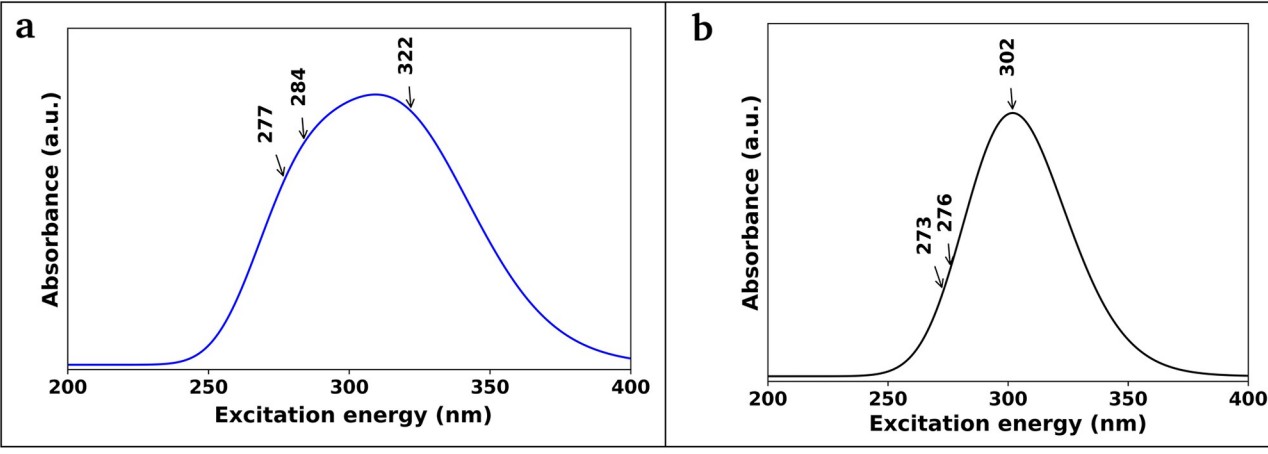

**Fig 9. Calculated ultraviolet-visible spectra.** (a) caffeic acid and (b) *p*-coumaric acid.

C5 in compound CA and C4 in compound pCA have the largest positive charge among all the carbon atoms, they are targets of nucleophilic attacks on the molecules, while C7 in CA and C5 in pCA carry the most significant negative charge among all the carbon atoms, they are targets of electrophilic attacks.

**Natural bond orbital.** The NBO analysis is useful for studying intramolecular bonding and the interactions among bonds which provides knowledge to investigate the charge transfer in molecular systems [82]. In NBO, donor-acceptor interactions involves the computation of

**Table 8. Mulliken and natural charges of all the atoms in caffeic acid and *p*-coumaric acid molecules.**

| Atoms of CA | Mulliken charges for CA | Natural charges for CA | Atoms of pCA | Mulliken charges for pCA | Natural charges for pCA |
|---|---|---|---|---|---|
| O1 | -0.235323 | -0.66895 | O1 | -0.224004 | -0.66277 |
| O2 | -0.301401 | -0.69844 | O2 | -0.184962 | -0.69039 |
| O3 | -0.182373 | -0.69008 | O3 | -0.316991 | -0.61573 |
| O4 | -0.314204 | -0.61374 | C4 | 1.366332 | -0.12625 |
| C5 | 1.767221 | -0.10240 | C5 | -1.296883 | -0.14217 |
| C6 | -0.034602 | -0.19797 | C6 | 0.366763 | -0.13955 |
| C7 | -1.367165 | -0.17498 | C7 | -0.131201 | -0.27419 |
| C8 | -0.291651 | 0.27624 | C8 | -0.312098 | -0.24974 |
| C9 | -0.186813 | 0.26449 | C9 | -0.523021 | 0.33329 |
| C10 | -0.108860 | -0.25855 | C10 | -0.125532 | -0.09168 |
| C11 | -0.246887 | -0.09121 | C11 | -0.192074 | -0.31927 |
| C12 | -0.185658 | -0.31488 | C12 | 0.039700 | 0.76068 |
| C13 | 0.022175 | 0.76080 | H13 | 0.076835 | 0.20632 |
| H14 | 0.190800 | 0.22060 | H14 | 0.171772 | 0.20843 |
| H15 | 0.062039 | 0.20693 | H15 | 0.158971 | 0.20389 |
| H16 | 0.166759 | 0.20526 | H16 | 0.192310 | 0.22239 |
| H17 | 0.177185 | 0.21812 | H17 | 0.170720 | 0.21728 |
| H18 | 0.207869 | 0.20908 | H18 | 0.201172 | 0.20855 |
| H19 | 0.284108 | 0.48633 | H19 | 0.270553 | 0.46946 |
| H20 | 0.285530 | 0.48179 | H20 | 0.291637 | 0.48144 |
| H21 | 0.291252 | 0.48157 | - | - | - |

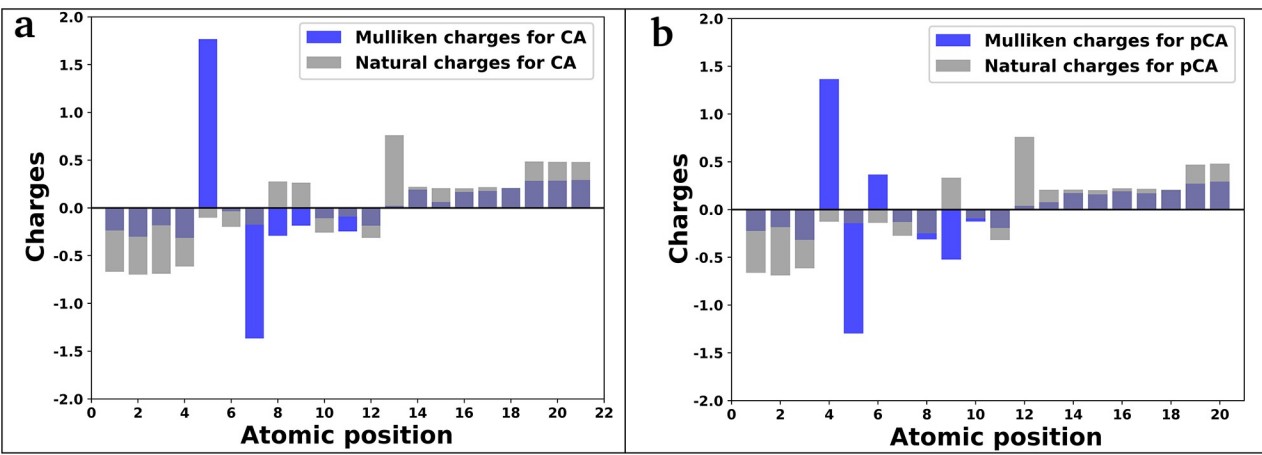

**Fig 10. Atomic charges calculated with Mulliken and NBO methods.** (a) caffeic acid and (b) *p*-coumaric acid.

the second order Fock matrix. The stabilization energy value depends on the difference between the energies of a particular acceptor and donor orbitals by expression (Eq 9) [83]

$$E(2) = q_i \frac{F^2(i, j)}{E(j) - E(i)} \tag{9}$$

where, $q_i$ is the donor orbital occupancy, $E(i)$ and $E(j)$ are diagonal elements and $F(i, j)$ is the off diagonal NBO Fock matrix elements.

The interactions from nonbonding donor orbitals LP(2)O3 to antibonding $\sigma^*$(O4-C13) leads to the highest stabilization of 41.24 kcal/mol for CA molecule. Again, LP(2)O4 to $\sigma$ (O3-C13), LP(2)O1 to $\sigma^*$(C6-C8), LP(2)O2 to $\sigma^*$(C9-C10) have stabilization energies 33.61 kcal/mol, 27.31 kcal/mol, 24.67 kcal/mol, respectively. In the case of $\pi$(C11-C12) to $\pi^*$ (O4-C13) gives moderate stabilization energy of 21.68 kcal/mol (S2 Table). The interactions from nonbonding donor orbitals LP(2)O2 to antibonding $\pi^*$(O3-C12) leads to the highest stabilization of 41.19 kcal/mol for pCA molecule. Again, LP(2)O3 to $\sigma^*$(O2-C12), LP(2)O1 to $\pi^*$ (C8-C9), $\pi$(C8-C9) to $\pi^*$(C4-C6) have stabilization energies of 33.56 kcal/mol, 27.55 kcal/mol, 24.48 kcal/mol, respectively (S3 Table).

## ADMET analysis

In accordance with Lipinski's rule of 5, oral drugs must obey a minimum of three out of four criteria: the molecular weight should not exceed 500 Da; the total count of hydrogen bond acceptors should not surpass 10; the total count of hydrogen bond donors should not exceed 5; and the octanol-water partition coefficient (LogP) should not exceed 5 (or MlogP$\leq$ 4.15) [30, 84]. The findings tabulated in S4 Table indicate that both compounds meet the criteria for Lipinski's rule of 5, suggesting their potential suitability as oral drug candidates. Again, important ADMET properties such as water solubility (LogS), which should exceed -5 and the topological polar surface area (TPSA), which must not exceed 140 Å$^2$ [31]. The estimated water solubility values for CA and pCA are -1.89 and -2.02, respectively. Additionally, the TPSA values for CA and pCA are 77.76 Å$^2$ and 57.53 Å$^2$, respectively. These data collectively demonstrate that all of these values fall within the acceptable range.

A common method of expressing toxic doses is through the LD50 (mg/kg body weight) values. Compound with LD50 value between 1000 mg/kg and 5000 mg/kg is classified as having

low or mild toxicity [85]. The LD50 values for CA and pCA are 2980 mg/kg and 2850 mg/kg, respectively, showing low toxicity in the compounds (S4 Table). The bioavailability radar evaluates a molecule's druglikeness based on six key physicochemical properties: lipophilicity, size, polarity, solubility, flexibility, and saturation. Each property is associated with a defined range on the radar plot, represented as a pink area. To be considered druglike, the molecule's radar plot must fall within this specified pink region [66]. The radar plots for both CA and pCA molecules covered the pink area across five physicochemical properties, except for saturation (S7 Fig).

The BOILED-Egg model predicts molecule absorption by the gastrointestinal tract in the white area and permeation of the blood-brain barrier in the yellow area based on lipophilicity (WLOGP) and polarity (TPSA) [86]. The CA molecule is located in the white region, suggesting a greater probability of gastrointestinal tract absorption. The pCA molecule is resided in the yellow region, implying an increased potential for passive permeation through the blood-brain barrier as well (S8 Fig).

## Conclusion

The findings of this study computationally explored the promising inhibitors of HDAC2 among the chosen anticancer phytocompounds. The docking scores demonstrated strong binding of the caffeic acid and *p*-coumaric acid compounds to the HDAC2 enzyme. Compound CA formed 5 H-bonds, whereas compound pCA formed 4 H-bonds with the active amino acids of HDAC2. The stronger binding efficacy of CA over pCA was attributed to the presence of an additional H-bond with Tyr297 residue of the enzyme by an extra oxygen atom in the former molecule. Small RMSD, RMSF, $R_g$, and SASA values of CA-HDAC2 and pCA-HDAC2 complexes, as well as the apo form of HDAC2, demonstrated conformational stability during the 200 ns MD simulations. Stable protein-ligand interactions of the complexes were also evidenced by results of MM/GBSA calculations.

The small values of HOMO and LUMO energy gaps, along with favourable values of dipole moment, chemical potential, chemical hardness, and electrophilicity, collectively indicated the reactive nature of the molecules using DFT calculations. Both the phenolic acid molecules under investigation further demonstrated good physicochemical and pharmacokinetic properties with no toxicity. In summary, both the compounds exhibited promising potential as drug against colon cancer, with CA as a more favourable candidate compared to pCA. However, it is essential to validate these findings through further preclinical experiments.

## Supporting information

**S1 Fig. Poses of co-crystalline ligand before (green) and after (red) docking at a binding site for calculation of RMSD using DockRMSD program.**
(TIFF)

**S2 Fig. Interactions of compounds with active amino acid residues of the protein.** (a) CA-HDAC2 and (b) pCA-HDAC2 complexes prepared by LigPlot+ V.2.2 program.
(TIFF)

**S3 Fig. RMSD of the heavy atoms of the ligand relative to the protein backbone for CA-HDAC2 and pCA-HDAC2 complexes.**
(TIFF)

**S4 Fig. The contributions of binding energy by actives residues and ligand.** (a) CA-HDAC2 and (b) pCA-HDAC2 complexes.
(TIFF)

**S5 Fig. Number of H-bonds with time of simulation for CA-HDAC2 complex and pCA-H-DAC2 complex.**
(TIFF)

**S6 Fig. Crystal structures depicting interactions between compounds and active amino acid residues of the HDAC2 protein at various frames.** (a) CA-HDAC2 and (b) pCA-H-DAC2 complexes. The 2D diagrams illustrate interactions of the compounds with active amino acid residues at the 2000[th] frame of MD simulations for (c) CA-HDAC2 and (d) pCA-HDAC2 complexes.
(TIFF)

**S7 Fig. Bioavailability Radar diagram for the druglikeness at single glance.** (a) CA molecule and (b) pCA molecule.
(TIFF)

**S8 Fig. BOILED-Egg for the evaluation of HIA and BBB in function of the position of the molecules in the WLOGP with TPSA of both CA and pCA molecules.**
(TIFF)

**S1 Table. Binding energy of some selected biomolecules for virtual screening using pyRx.**
(PDF)

**S2 Table. Second order perturbation theory analysis of Fock matrix on NBO basis for CA molecule using B3LYP/6-311++G(d,p).**
(PDF)

**S3 Table. Second order perturbation theory analysis of Fock matrix on NBO basis for pCA molecule using B3LYP/6-311++G(d,p).**
(PDF)

**S4 Table. Results of ADME and toxicity parameters for caffeic acid and p-coumaric acid.**
(PDF)

## Acknowledgments

We thank Prof. Dr. Rajendra Parajuli and Asst. Prof. Pitambar Shrestha from Amrit Science College (ASCOL), Tribhuvan University (TU), Kathmandu, Nepal, for providing access to Gaussian software.

## Author Contributions

**Conceptualization:** Madan Khanal.

**Data curation:** Madan Khanal, Arjun Acharya, Rajesh Maharjan, Kalpana Gyawali, Tika Ram Lamichhane, Hari Prasad Lamichhane.

**Formal analysis:** Madan Khanal.

**Investigation:** Madan Khanal.

**Methodology:** Madan Khanal.

**Project administration:** Madan Khanal.

**Resources:** Madan Khanal.

**Software:** Madan Khanal.

**Supervision:** Hari Prasad Lamichhane.

**Visualization:** Madan Khanal.

**Writing – original draft:** Madan Khanal.

**Writing – review & editing:** Madan Khanal, Arjun Acharya, Rajesh Maharjan, Kalpana Gyawali, Rameshwar Adhikari, Deependra Das Mulmi, Tika Ram Lamichhane, Hari Prasad Lamichhane.

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
