## [Decision Letter · Decision Letter 0]

7 May 2024

PONE-D-24-13945Identification of potent inhibitors of HDAC2 from herbal products for the treatment of colon cancer: Molecular docking, molecular dynamics simulation, MM/GBSA calculations, DFT studies, and pharmacokinetic analysisPLOS ONE

Dear Dr. Khanal,

Thank you for submitting your manuscript to PLOS ONE. After careful consideration, we feel that it has merit but does not fully meet PLOS ONE’s publication criteria as it currently stands. Therefore, we invite you to submit a revised version of the manuscript that addresses the points raised during the review process.

We look forward to receiving your revised manuscript.

Kind regards,

Ahmed A. Al-Karmalawy, PhD

Academic Editor

PLOS ONE

Journal Requirements:

"The authors declare no conflict of interest."

**Reviewers' comments:**

Reviewer's Responses to Questions

**Comments to the Author**

1. Is the manuscript technically sound, and do the data support the conclusions?

Reviewer #1: Partly

Reviewer #2: Yes

Reviewer #3: Partly

2. Has the statistical analysis been performed appropriately and rigorously? 

Reviewer #1: N/A

Reviewer #2: N/A

Reviewer #3: No

3. Have the authors made all data underlying the findings in their manuscript fully available?

Reviewer #1: Yes

Reviewer #2: Yes

Reviewer #3: Yes

4. Is the manuscript presented in an intelligible fashion and written in standard English?

Reviewer #1: Yes

Reviewer #2: Yes

Reviewer #3: Yes

5. Review Comments to the Author

Reviewer #1: Recommendation: This paper is publishable subject to major revisions.

Comments:

In the manuscript titled “Identification of potent inhibitors of HDAC2 from herbal products for the treatment of colon cancer: Molecular docking, molecular dynamics simulation, MM/GBSA calculations, DFT studies, and pharmacokinetic analysis” the authors investigated the effectiveness of two phytocompounds, caffeic acid and ρ-coumaric acid as drugs for the treatment of colon cancer by exploring their binding ability to histone deacetylase 2 (HDAC2) enzyme employing molecular docking, molecular dynamics and DFT analyses. However, the manuscript is not publishable in its current form and requires a major revision. In this regard, I summarize my review comments below:

1. In the material and methods section, under molecular docking, the authors mentioned that “Two compounds showing the least binding energies from above screening were docked…”. If the compounds have the least binding energy, why did the authors choose them to study docking?

2. In the molecular dynamics simulation, from Figure 3a, it is noticed that for ρ-coumaric acid the RMSD started to diverge after 90 ns simulation. Please explain what the reason is for this divergence. It seems that 100 ns simulation is not enough for ρ-CA bound enzyme complex, and it is suggested to increase the simulation time.

3. In Table 2, under the binding residues, Asp170 was listed. However, in Table 3, under active amino acids, Asp93 was listed. The authors should check this and comment on why two different Asp residues were obtained.

4. The authors have given two statements, “The maximum fluctuation in RMSF for both complexes and apo form is occurred at the residues 198, 199, and 200 (Figure 4).” and “Again, among the active residues, Asp93 residue fluctuates with the maximum RMSF value for both the complexes as well as apo form (Table 3).” The reasons behind these fluctuations need to be explained and also check if the Asp 93 residue number is correct or not.

5. Did the authors replicate the simulations and observe “In CA-HDAC2 complex, the highest number of H-bonds formed is 4 during MD simulations. Between 65 ns and 78 ns, majority of conformations show 3 H-bonds”? If not, it is advised to replicate the simulations thrice and check whether the statement holds for the other two replications or not.

6. The statement “Within the active residues of HDAC2, MET24 has the lowest binding energy….” is not consistent with Figure 5e, f as in those figures MET24 and TYR18 have the highest binding energy for CA and ρ-CA, respectively.

7. The statement “The LD50 values for CA and pCA are 2980 mg/kg and 2850 mg/kg, respectively, showing absence of toxicity in these compounds” needs to be modified, as the values imply low toxicity.

8. The overall picture quality of the figures needs to be improved as it is hard to get the values, and residue names from the figures.

9. “The average Rg values of Cα atoms for the apo form (1.99 ± 0.01 nm), CA-HDAC2 complex (1.98 ± 0.01 nm), and the pCA-HDAC2 complex (1.98 ± 0.01 nm) are nearly identical.” Maintain consistency of unit between figure (figure 3c) and text.

10. “The average value of RMSF for apo form is 0.72 ± 0.43 Å and for CA-HDAC2 and CA-HDAC2 complexes…”. The second CA-HDAC2 should be ρ-CA.

11. Table 5 heading, place the description according to the table.

Reviewer #2: The manuscript presents a computational analysis of phytocompounds with anticancer properties by inhibiting HDAC2. The analysis includes docking, MD simulation, MM/GBSA calculations, DFT studies, and pharmacokinetic studies. The manuscript is well-written, and I recommend publishing it with the following revisions:

1. The authors mentioned that among the seventeen phytochemicals, the molecules caffeic acid and p-coumaric acid showed high binding efficacy with HDAC2. However, upon reviewing Table 1S, I noticed that Epipodophyllotoxin and Ferulic acid also showed the same binding efficacy with p-coumaric acid. Could the authors explain why they proceeded with only the first two molecules? Furthermore, based on what criteria did the authors decide not to proceed with Epipodophyllotoxin and Ferulic acid for other computational analyses?

2. The authors should discuss the stability of the observed interactions in the docked pose between the ligands and HDAC2 in the MD simulation. Additionally, it would be beneficial to explain how long these interactions sustained during the simulation period.

Reviewer #3: Overall, the manuscript has tried to use the most available computational methods to study the two compounds. However, there are many nuances where it could have been more helpful and exciting to add to the current trend in computer-aided drug design, where a few different scaffolds were studied and then compared based on the difference in their molecular features. The readers are just introduced to two highly similar caffeic acid and p-coumaric acid, which perform comparably across all quantifications as expected. I am concerned about the lack of experimental validation, particularly because caffeic acid is more reactive due to the presence of two hydroxyl groups, making it susceptible to oxidation and other chemical reactions. However, this may be helpful as it will help scavenge all ROS species in the stressed cells. But could it cause potential cytotoxicity and further damage? The Bioavailability of caffeic acid has also been questioned recently due to its rapid metabolism. The overall comparison of both compounds in the text, with similar values all along, lacks a strong agreement with caffeic acid being a strong compound to treat cancer.

6. PLOS authors have the option to publish the peer review history of their article (what does this mean?). If published, this will include your full peer review and any attached files.

Reviewer #1: No

Reviewer #2: No

Reviewer #3: **Yes**

---

## [Author Response · Author response to Decision Letter 0]

21 Jun 2024

Jun 21, 2024

To

The Reviewers

PLOS ONE

Subject: Response to Academic Editor and Reviewers' Comments

Dear Academic Editor and Reviewers,

We are submitting our revised manuscript entitled “Identification of potent inhibitors of HDAC2 from herbal products for the treatment of colon cancer: Molecular docking, molecular dynamics simulation, MM/GBSA calculations, DFT studies, and pharmacokinetic analysis” (Manuscript id: PONE-D-24-13945) for publication in PLOS ONE. We appreciate the time and effort that the reviewers have invested in evaluating our work. Their insightful comments and suggestions have been invaluable in improving the quality of our manuscript. We have highlighted our responses to the reviewer's comments in the revised manuscript with yellow color. Below, we provide a detailed response to each of the points raised.

We hope that the revisions and responses meet your expectations and we look forward to your positive feedback. I hereby declare that all the authors involved in this work have approval for the submission process and the authors have declared that no competing interests exist.

Sincerely,

Madan Khanal

Assistant Professor, Tribhuvan University

Kathmandu, Nepal

Email: madan.khanal@bumc.tu.edu.np

Academic Editor:

1. Introduction :

a. In the introduction, highlight a few previous studies illustrating the importance of computational tools in studying anticancer, specifically plant-derived phenolic compounds.

Response: Thank you for the suggestion. We have highlighted some previous studies about the importance of computational tools in studying anticancer properties using plant-derived phenolic compounds in third paragraph of the “Introduction” section.

b. Describe the secondary structure composition of HDAC2 as the author talks about the importance of the residues in the binding site. Where does the active site lie?

Response: Thank you for your suggestion. The information of residues of active site and secondary structure composition of HDAC2 are addressed in the first paragraph of the “Introduction” section.

2. Section 2.1 Molecular docking : 

a. The active amino acid numbering of the protein was compared between RCSB PDB and SWISS-MODEL, revealing differences in the numbering sequence

i. What does the author mean by the difference in the number sequence and the number of missing residues, and in which region of the protein?

Response: Thank you for insightful comments. In RCSB PDB of HDAC2 enzyme (PDB ID: 3MAX) starts from Ala12 and ends on Leu378. In SWISS-MODEL, the protein starts from Ala1 and ends on Leu367. This is due to differences in reference sequences; however, there are no missing residues. This is also incorporated in the first paragraph of subsection “Molecular docking” under the section “Materials and Method”. 

ii. The total number of residues in the protein should be mentioned. 

Response: Total number of residues in the protein are 367, which is now mentioned in the first paragraph of subsection “Molecular docking” under the section “Materials and Method”.

iii. Highlight the residues missing in the crystal structure 

 Response: There are no any missing residues in the crystal structure.

iv. Which template was used as a template to build the homology model 

Response: During refinement, the template was found using HHblits with a global model quality estimate (GMQE) score of 0.99, derived from X-ray diffraction at a resolution of 1.66 Å, representing the monomeric form of the protein structure. This is also included in the first paragraph of subsection “Molecular docking” under the section “Materials and Method”.

b. The bioactive molecules having anticancer properties were chosen from the literature

i. How many compounds were incorporated? Mention the dataset here.

Response: We have selected seventeen bioactive compounds and these are tabulated in “S2 Table” with binding energy scores against HDAC2.

ii. How were the anticancer compounds filtered from PubChem? What was the rationale behind creating the data? Describe a particular scaffold to add to the information.

Response: We selected the anticancer compounds from literature sources and their structures were downloaded from PubChem online database. The anticancer compounds were not filtered from PubChem. These compounds are commonly found in the various plant-based foods and extensively studied for their potential anticancer properties.

c. Where is the grid box located? The author should highlight and describe a specific region/active site. Was the whole protein used for docking? Clarify. 

Response: Docking was done within the grid points 60 x 60 x 60 along x, y, and z directions, respectively, at grid center x = 67.987, y = 31.384, and z = 3.889 with grid spacing 0.375 Å. The active site are lipophilic tube formed by amino acids Gly143, Phe144, His172, Phe199, and Leu265 with foot pocket adjacent to lipophilic tube formed by Tyr18, Met24, Phe103, and Leu133. Docking was done on the active site of the protein. Whole protein was not used for the docking process. This details is incorporated in the second paragraph of subsection “Molecular docking” under the section “Materials and Method”.

3. Section 2.1 Molecular Dynamics simulations :

a. Which force field was employed for the small molecules?

Response: Thank you for the concern. In molecular dynamics simulations, Chemistry at HARvard Macromolecular Mechanics (CHARMM) force filed was employed. The CHARMM27 parameter set was used, and it was parameterized using the SwissParam online tool.

b. Mention the length of the total simulation.

Response: The total length of simulation is 200 nanoseconds (ns) which is included in the subsection “Molecular dynamics simulations” under “Materials and Methods” section.

c. The strength of NaCl used to neutralize the system.

Response: In the molecular dynamics (MD) simulation, the concentration of 0.1 M (molar) NaCl was used to neutralize the system. This is also added in the subsection “Molecular dynamics simulations” under “Materials and Methods” section.

d. How was the apo system prepared?

Response: To prepare the apo system (the protein without a bound ligand) of HDAC2, we refined the downloaded protein (PDB ID: 3MAX) using SWISS-MODEL. Then, the bound ligand in the protein was removed, and polar hydrogen atoms and Kollman charges were added.

e. Are any terminal patches applied to the protein to neutralize them?

Response: The polar hydrogen atoms and the Kollman charges were added using AutoDock Tools (ADT) before molecular docking. During molecular dynamics simulations, we have not applied any terminal patches to the protein.

f. Was the crystal water retained during the simulations?

Response: Yes. All crystal water molecules were kept during the simulations employing TIP3P water model.

g. How were the results from MD simulations analyzed? Any clustering method used to study the docked complexes?

Response: The results of MD simulations were analyzed by trajectory analysis for the structural and dynamic properties of system over time. This involves RMSD, RMSF, Rg, SASA, H-bonded interaction patterns etc. We used free energy calculations (MM/GBSA approach) to estimate the binding free energy of the protein-ligand complexes. We also used PyMOL 2.5.2, LigPlot+ V.2.2, and BIOVIA Discovery Studio visualizer V21.1.0.20298 to visualize and analyze MD trajectories.

4. MM/GBSA approach

a. What are the main contributors to the common binding energy across various interactions, specifically in GbindvdW, GbindLipo, and GbindCoulomb energies?

Response: Thank you for your concern. The main contributors of different interactions are further explained in the subsection “MM/GBSA approach” under the “Materials and Methods” section.

b. Does MM/GBSA calculation consider the water molecule in the active site? If the authors had tried these calculations without crystal water and had compared the results, would it have been worse if the water had been included?

Response: In gmx_mmpbsa, explicit water molecules are removed, and their effects are implicitly included using solvation model like PB or GB methods. The MD simulation of the protein-ligand complex was performed using an explicit solvent model and all the solvent molecules and charged ions were deleted from each MD snapshot, and the implicit PBSA or GBSA solvent model was used to evaluate the solvation energy.

c. How does one validate the precision of data from MM/GBSA calculation while comparing two compounds? If employed, the authors should briefly describe a method. 

Response: We used the following methodology for validation of the results.

Simulation Conditions: Both compounds were subjected to MD simulations under identical conditions using the V-rescale thermostat and Parrinello-Rahman barostat. The temperature was maintained at 300 K and pressure at 1 bar for a duration of 200 ns with a 1 fs time step.

Trajectory Recording: Trajectories were recorded every 0.1 ns, ensuring a consistent data set for subsequent analysis. Using the same force field (AMBER) and parameters for both compounds. Running simulations under identical conditions (e.g., temperature, pressure, simulation time). Applying the same MM/GBSA calculation settings (e.g., dielectric constants, solvent model).

RMSD Analysis: We plotted the Root Mean Square Deviation (RMSD) for both compounds over the entire simulation period. This allowed us to confirm that both systems had equilibrated and reached a stable conformational state.

RMSF, Radius of gyration, SASA, and H-bond Interactions: We analyzed these parameters to understand the conformational stability and flexibility of the compounds, ensuring that the observed binding free energies were representative of stable conformations.

We also calculated the average binding free energy and standard deviation for each compound. We included error bars in the final reported binding free energy values to represent the uncertainty the calculations. The binding poses of the two compounds were visualized and compared to ensure that the binding modes were consistent and meaningful.

d. How do the authors address the statistical convergence of results from MM/GBSA?

Response: The MM/GBSA method has been investigated with the aim of achieving a statistical precision of 1 kJ/mol for the results. High number of snapshots, performing visual and quantitative analyses (RMSD analysis) to confirm that the system has reached equilibrium, the moving average plots of the binding energy as a function of number of frames, small standard error of the mean indicates better convergence, tools like AMBER, GROMACS with g_mmpbsa have built-in features to aid in convergence analysis.

The results from the last 50 ns and last 20 ns are quite similar. The whole 100 ns result is also consistent with these segments, indicating that the inclusion of the entire simulation did not significantly affect the free energy estimate.

e. Were any rotamer search techniques involved during free energy estimation?

Response: Thank you for the concern. Any rotamer search techniques were not involved during free energy estimation.

5. Results: Molecular dynamics

i. The authors should elaborate on the docked ligand's RMSD compared to the crystal structures and discuss any deviations in interactions with the active site residues. The simulation clustered data also gives a more detailed insight into which bonds stay stable during the simulation.

Response: Thank you for the suggestion. We have elaborated the docked ligand's RMSD with crystal structure along with different interactions in the first paragraph of subsection “Molecular dynamics simulations” under the section “Results and Discussion”.

ii. How many simulation replicates were run for the analysis of each compound? If the authors ran only one, how would they emphasize and ensure the simulations were converged?

Response: Initially, we had run two simulation replicates of duration 100ns time. Now, we have run two simulation replicates of duration 200ns for the analysis of each compound. After two simulations, we found almost similar type results with small change in the values of RMSD, RMSF, Rg, SASA, H-bonds formation, and MM/GBSA binding free energies during 200ns simulation time period. We got negative and almost same MM/GBSA binding energy values for each replication of MD simulations. After then, we have analyzed for single simulations for each compound complexes as well as apo form of the HDAC2 protein to write the manuscript.

iii. The average Rg values of Cα atoms for the apo form (1.99 ± 0.01 nm), CA-HDAC2 complex (1.98 ± 0.01 nm), and the pCA-HDAC2 complex (1.98 ± 0.01 nm) are nearly identical. The HDAC2 protein is significantly stable in the presence of CA and pCA molecules during the simulation period with low average values of Rg.

1. The authors claim that HDAC2 protein is more stable in the presence of CA than pCA. However, the Rg values are exactly identical. Even when compared to apo, the protein does not show much instability in the absence of either of these compounds. This rationale seems weak, and the authors should suggest a stronger point to comment on the stability of the complex.

Response: Thank you for the suggestion. The Rg values for all three cases are almost identical and low, and hence both the complexes as well as apo form of HDAC2 are stable.

6. Results: MM/GBSA

a. Within the active residues of HDAC2, MET24 has the lowest binding energy of -2.44 ± 1.00 kcal/mol in CA-HDAC2 complex and TYR18 has the lowest binding energy of -1.73 ± 0.75 kcal/mol in pCA-HDAC2 complex

i. Inconsistency is residue annotation “Tyr18.”

Response: Thank you for the review. This is corrected in the revised manuscript to make consistency in residue annotation. We corrected as “Tyr18” and “Met24”.

b. The average value of RMSF for apo form is 0.72 ± 0.43 A and ˚ for CA-HDAC2 and CA-HDAC2 complexes are 0.83 ± 0.36 A and 0.83 ˚ ± 0.51 A, respectively.

i. Did the authors mean CA-HDAC2 and pCA-HDAC2?

Response: Thank you for the careful review. This typo is corrected in the revised manuscript. It is “CA-HDAC2 and pCA-HDAC2”.

7. Grammer inconsistencies in the text.

Response: Thank you for the suggestion. We have reviewed the manuscript and tried to mitigate the grammatical errors in the text.

8. Figure 1s Atomic notation 

Response: Name of co-crystalline ligand is N-(4-aminobiphenyl-3-yl)benzamide and the atomic notation is (C19H16N2O) and atomic notation are also shown in the figure.

9. Figure 2: The figures' quality is hazy, so they should be saved in TIFF format to achieve the maximum quality for publication images. 

Response: Thank you for your suggestion. We have improved the figures' quality by remaking some of them and saving them in TIFF format to ensure maximum quality for publication.

10. Figure 3, panel A, RMSD of complex / ligand/ protein backbone? Kindly update the label on the y-axis appropriately.

Response: Thank you for the suggestion. The “RMSD of protein’s backbone” is updated in y-axis of the figure.

11. Figure 3, panel E, SASA of what needs to be mentioned and updated. 

Response: Thank you for the suggestion. We have calculated SASA of protein and is mentioned in the figure.

12. Figure 4, Describe the difference in RMSF between all three system set-ups at residue number ~198 (spike region ~5 Angstrom). The authors should elaborate on where this residue lies and why we see a significantly low RMSF in CA during the simulations. 

Response: Thank you for the comment. This is elaborated in the fourth paragraph of subsection “Molecular dynamics simulations” under the section “Results and Discussion”.

13. A figure illustrating the alignment of the docked compound to the sampled conformations from MD simulation and crystal structure could be meaningful.

Response: Thank you for the suggestion. Structure of 2000th frame which is included in the revised manuscript. The interacting amino acids with both the compounds before MD simulations, after molecular docking are almost same after the 2000th frame MD simulations. This shows the correspondence between the docked compounds and the conformations obtained from MD simulations and the crystal structure. This is elaborated in the sixth p

---

## [Decision Letter · Decision Letter 1]

8 Jul 2024

Identification of potent inhibitors of HDAC2 from herbal products for the treatment of colon cancer: Molecular docking, molecular dynamics simulation, MM/GBSA calculations, DFT studies, and pharmacokinetic analysis

PONE-D-24-13945R1

Dear Dr. Khanal,

We’re pleased to inform you that your manuscript has been judged scientifically suitable for publication and will be formally accepted for publication once it meets all outstanding technical requirements.

Kind regards,

Ahmed A. Al-Karmalawy, PhD

Academic Editor

PLOS ONE

Reviewers' comments:

Reviewer's Responses to Questions

**Comments to the Author**

1. If the authors have adequately addressed your comments raised in a previous round of review and you feel that this manuscript is now acceptable for publication, you may indicate that here to bypass the “Comments to the Author” section, enter your conflict of interest statement in the “Confidential to Editor” section, and submit your "Accept" recommendation.

Reviewer #1: All comments have been addressed

Reviewer #2: All comments have been addressed

Reviewer #3: All comments have been addressed

2. Is the manuscript technically sound, and do the data support the conclusions?

Reviewer #1: Yes

Reviewer #2: Yes

Reviewer #3: Yes

3. Has the statistical analysis been performed appropriately and rigorously? 

Reviewer #1: N/A

Reviewer #2: N/A

Reviewer #3: Yes

4. Have the authors made all data underlying the findings in their manuscript fully available?

Reviewer #1: Yes

Reviewer #2: Yes

Reviewer #3: Yes

5. Is the manuscript presented in an intelligible fashion and written in standard English?

Reviewer #1: Yes

Reviewer #2: Yes

Reviewer #3: Yes

6. Review Comments to the Author

Reviewer #1: All the concerns raised in previous revision have been addressed well and the manuscript is acceptable in it's current form.

Reviewer #2: Dear Editor,

I have reviewed the revised manuscript and found that the authors have thoroughly addressed all my previous comments and concerns. The revisions have strengthened the manuscript, and I now recommend it for publication.

Reviewer #3: Overall, the authors suggest a caffeic acid(CA) and p-coumaric acid(pCA) compounds, as possible HDAC2 inhibitors. Because of an extra H-bond with the Tyr residue of the enzyme, CA demonstrated greater binding effectiveness, represented by both conformational stability and stable protein-ligand interactions. With CA being a more promising contender, both compounds showed good physicochemical and pharmacokinetic features without toxicity.

However, experimental findings to support the stability and toxicity of the computational results would add more meaning to the comparison between compounds with a highly similar scaffold. Additional preclinical research is required to confirm these results.

All comments have been duly addressed.

7. PLOS authors have the option to publish the peer review history of their article (what does this mean?). If published, this will include your full peer review and any attached files.

Reviewer #1: No

Reviewer #2: No

Reviewer #3: No

---

## [Editor Report · Acceptance letter]

12 Jul 2024

PONE-D-24-13945R1 

PLOS ONE

Dear Dr. Khanal, 

I'm pleased to inform you that your manuscript has been deemed suitable for publication in PLOS ONE. Congratulations! Your manuscript is now being handed over to our production team.

Kind regards, 

on behalf of

Associate Professor Ahmed A. Al-Karmalawy 

Academic Editor

PLOS ONE